# Structural mechanisms of phospholipid activation of the human TPC2 channel

Ji She[1,2], Weizhong Zeng[1,2,3], Jiangtao Guo[4,5], Qingfeng Chen[1,2,3], Xiao-chen Bai[2,6]*, Youxing Jiang[1,2,3]*

[1]Department of Physiology, University of Texas Southwestern Medical Center, Dallas, United States; [2]Department of Biophysics, University of Texas Southwestern Medical Center, Dallas, United States; [3]Howard Hughes Medical Institute, University of Texas Southwestern Medical Center, Dallas, United States; [4]Department of Biophysics, Zhejiang University School of Medicine, Hangzhou, China; [5]Department of Pathology of Sir Run Run Shaw Hospital, Zhejiang University School of Medicine, Hangzhou, China; [6]Department of Cell Biology, University of Texas Southwestern Medical Center, Dallas, United States

*For correspondence:
Xiaochen.Bai@UTSouthwestern.
edu (X-B);
youxing.jiang@utsouthwestern.
edu (YJ)

Competing interests: The authors declare that no competing interests exist.

**Abstract** Mammalian two-pore channels (TPCs) regulate the physiological functions of the endolysosome. Here we present cryo-EM structures of human TPC2 (HsTPC2), a phosphatidylinositol 3,5-bisphosphate ($PI(3,5)P_2$)-activated, $Na^+$ selective channel, in the ligand-bound and apo states. The apo structure captures the closed conformation, while the ligand-bound form features the channel in both open and closed conformations. Combined with functional analysis, these structures provide insights into the mechanism of $PI(3,5)P_2$-regulated gating of TPC2, which is distinct from that of TPC1. Specifically, the endolysosome-specific $PI(3,5)P_2$ binds at the first 6-TM and activates the channel – independently of the membrane potential – by inducing a structural change at the pore-lining inner helix (IS6), which forms a continuous helix in the open state but breaks into two segments at Gly317 in the closed state. Additionally, structural comparison to the voltage-dependent TPC1 structure allowed us to identify Ile551 as being responsible for the loss of voltage dependence in TPC2.
DOI: https://doi.org/10.7554/eLife.45222.001

## Introduction

Two-pore channels (TPCs) belong to the voltage-gated ion channel superfamily and function as a homodimer, with each subunit containing two homologous *Shaker*-like 6-TM repeats. TPC1 and TPC2 represent two major subfamilies of mammalian TPC channels, and have been subjected to extensive studies (*Grimm et al., 2017*; *Patel, 2015*). TPC1 and TPC2 are localized to the endolyso-somal membrane and control the ionic homeostasis of these acidic organelles. TPC functions have been shown to be associated with a plethora of physiological processes, including mTOR-dependent nutrient sensing (*Cang et al., 2013*), Ebola infection (*Sakurai et al., 2015*) and autophagy (*Fernández et al., 2016*; *Pereira et al., 2011*). Despite their physiological importance, the biophysical properties of mammalian TPC channels are still under debate. Initial studies suggested that TPCs mediate nicotinic acid adenine dinucleotide phosphate (NAADP)-dependent calcium release from the endolysosome (*Brailoiu et al., 2009*; *Calcraft et al., 2009*; *Zong et al., 2009*). However, several recent studies have demonstrated that mammalian TPCs are sodium-selective channels activated by the endolysosome specific lipid phosphatidylinositol 3,5-bisphosphate ($PI(3,5)P_2$) rather than NAADP (*Cang et al., 2013*; *Wang et al., 2012*). The plasma membrane-localized $PI(4,5)P_2$ cannot activate the TPC channels; thus, the lipid isoform-dependent gating property defines the compartment

specific channel activity of TPCs. Distinct from TPC2, mammalian TPC1 activation is also voltage-dependent, conferring electrical excitability to the endolysosome (*Cang et al., 2014*).

Our recent cryo-EM structure of mouse TPC1 (MmTPC1) provided crucial insights into the ion selectivity and gating mechanisms of mammalian TPC channels – particularly the structural basis of PI(3,5)P$_2$ and voltage-dependent gating of mammalian TPC1 (*She et al., 2018*). TPC2, however, shares low sequence identity to MmTPC1 (24%, *Figure 1—figure supplement 1*) and exhibits distinct physiological and gating properties. While TPC1 is widely expressed throughout the endolysosomal pathway, TPC2 is preferentially expressed in the late endosome/lysosome (*Brailoiu et al., 2009*; *Calcraft et al., 2009*); TPC2 has been shown to be associated with Parkinson's disease (*Hockey et al., 2015*) and blood vessel formation (*Favia et al., 2014*); TPC2-knockout mice are highly susceptible to non-alcoholic fatty liver hepatitis, the role of TPC1, however, remains unclear (*Grimm et al., 2014*). Additionally, unlike the voltage-dependent TPC1, TPC2 can be activated solely by PI(3,5)P$_2$ binding. Here we present the cryo-EM structures of human TPC2 (HsTPC2) in both the ligand-bound and apo states, which reveal the structural basis of phospholipid binding as well as its activation mechanism.

## Results and discussion

### Overall structure of HsTPC2

By replacing Leu11 and Leu12 on the N-terminal lysosomal targeting sequence with alanines, HsTPC2 can be over-expressed and trafficked to the plasma membrane in HEK293 cells (*Brailoiu et al., 2010*; *Jha et al., 2014*). This mutation (Leu11Ala/Leu12Ala) results in better protein expression and also allows for direct measurement of channel activity by patch-clamp recording of the plasma membrane. Therefore, this mutant construct was used and defined as the wild-type channel in our structural and functional studies.

The structures of HsTPC2 were determined in the presence and absence of PI(3,5)P$_2$ by single particle cryo-EM (*Figure 1*, *Figure 1—figure supplements 2*, *3* and *4* and *Figure 1—source data 1*). The apo structure of HsTPC2 represents the closed conformation and was determined to 3.5 Å. Interestingly, the single particles of HsTPC2 in the presence of PI(3,5)P$_2$ can be partitioned into two classes of particles, and 3D refinement yielded HsTPC2 structures in the open and closed conformations at 3.7 Å and 3.4 Å resolution, respectively. The 3:5 ratio of the particles between the open and closed conformations in the PI(3,5)P$_2$-bound structure of HsTPC2 likely represents the ligand efficacy of channel activation. Other than the presence or absence of ligand density, the structures of the apo closed and the ligand-bound closed states are virtually identical with a main-chain RMSD of 0.46 Å; we, therefore, will use the ligand-bound structures in most of the discussion. It is worth noting that the observation of both the closed and open conformations of a ligand-gated channel in the presence of ligand has been observed in other channel structural studies using single particle cryo-EM such as the Slo2 channel (*Hite and MacKinnon, 2017*).

Like other TPC channels, HsTPC2 functions as a dimer with each subunit containing two homologous six-transmembrane domains (6-TM I and 6-TM II) (*Figure 1*). Following the same nomenclature as other voltage-gated channels, we labeled the six transmembrane helices within each 6-TM domain as IS1-S6 and IIS1-S6, respectively (*Figure 1A*). Like most voltage-gated ion channels, the transmembrane region of HsTPC2 is domain swapped, with the S1-S4 voltage-sensing domain (VSD) from one 6-TM interacting with the S5-S6 pore domain from the neighboring 6-TM (*Figure 1B*). Despite sharing low sequence identity, the overall structures of HsTPC2 and MmTPC1 can be well aligned (*Figure 1C*). Similar to MmTPC1, the linker between the two 6-TMs adopts an EF-hand domain structure but with two EF-hand motifs (EF-1 and EF-2) lacking the essential acidic residues for Ca$^{2+}$ coordination and an E1 helix extending from the C-terminus of the exceptionally long IS6 helix (*Figure 1A,B* and *Figure 1—figure supplement 1*). A major difference between the two channels is that HsTPC2 lacks the long C-terminal region that forms a horseshoe-shaped structure around the EF-hand domain in MmTPC1 (*She et al., 2018*).

### Ion conduction pore of HsTPC2

The HsTPC2 ion conduction pore consists of S5, S6 and two pore helices (*Figure 1A*). In the channel dimer, two sets of filter residues enclose the central ion pathway with different dimensions

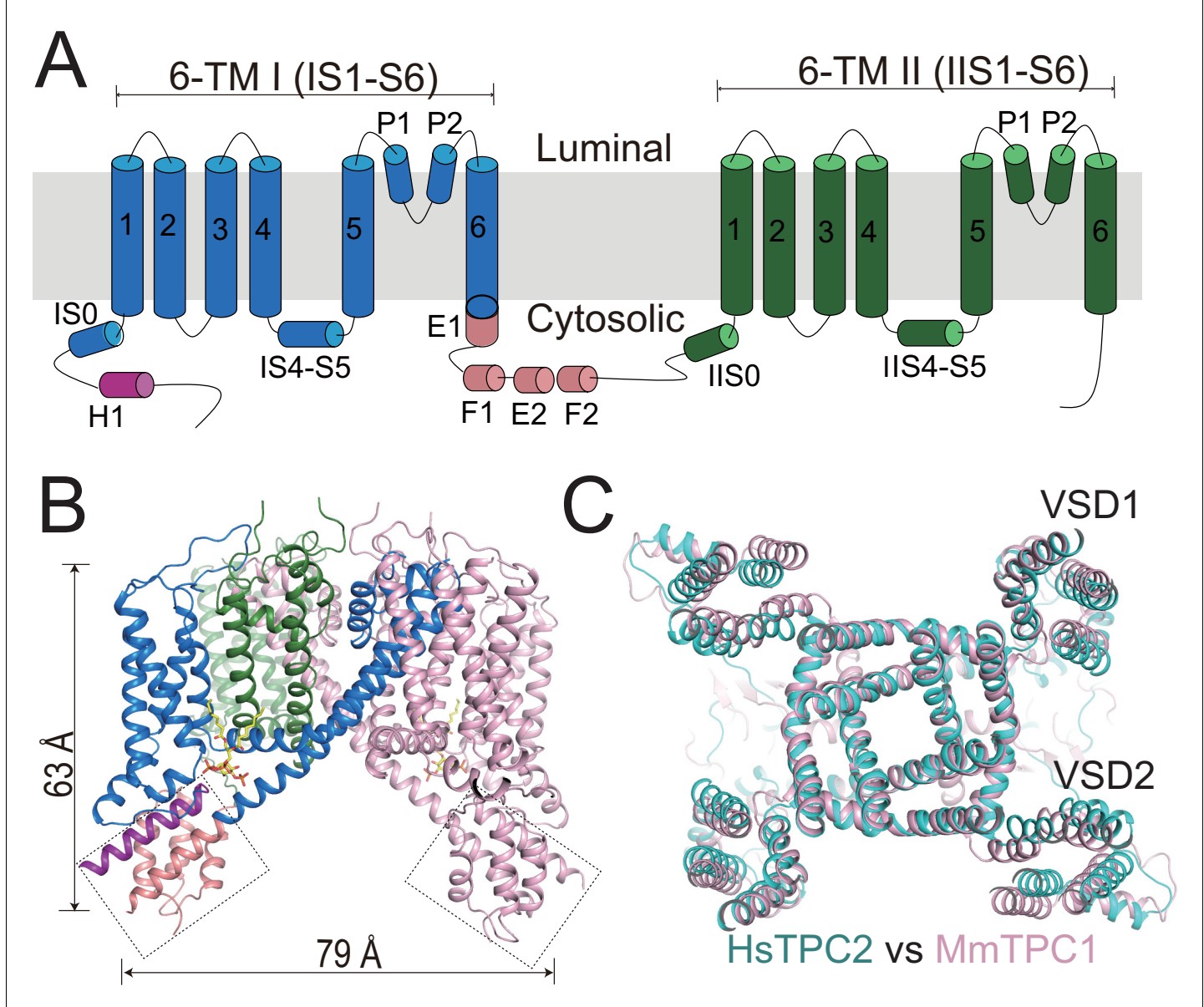

**Figure 1.** Overall structure of HsTPC2. (**A**) Topology and domain arrangement of an HsTPC2 subunit. (**B**) Cartoon representation of HsTPC2 with one subunit colored as that in (**A**) and another in pink. PI(3,5)P$_2$ are shown as yellow sticks. The two EF-hand domains are boxed. (**C**) Overall structural alignment of HsTPC2 (cyan) and MmTPC1 (pink).

DOI: https://doi.org/10.7554/eLife.45222.002

The following source data and figure supplements are available for figure 1:

**Source data 1.** Cryo-EM data collection and model statistics.
DOI: https://doi.org/10.7554/eLife.45222.007

**Figure supplement 1.** Sequence alignment of HsTPC2, MmTPC1 and AtTPC1.
DOI: https://doi.org/10.7554/eLife.45222.003

**Figure supplement 2.** Structure determination of the PI(3,5)P$_2$-bound HsTPC2.
DOI: https://doi.org/10.7554/eLife.45222.004

**Figure supplement 3.** Structure determination of the apo HsTPC2.
DOI: https://doi.org/10.7554/eLife.45222.005

**Figure supplement 4.** Sample EM density maps (blue mesh) for HsTPC2.
DOI: https://doi.org/10.7554/eLife.45222.006

(*Figure 2A*): filter I, with the sequence Thr271-Ala272, lines the pathway using main-chain backbone carbonyls with atom-to-atom cross distances of larger than 7 Å; filter II, with the sequence Val652-Asn653-Asn654, utilizes side chains to generate a much narrower pathway – with the two Asn653 residues forming the narrowest constriction point and playing the central role in determining $Na^+$ selectivity (*Guo et al., 2017b*; *She et al., 2018*). In sharing identical filter sequence and structure, mammalian TPC1 and TPC2 should utilize the same structural mechanism to achieve $Na^+$ selectivity as was demonstrated in our previous study (*She et al., 2018*). In plant TPC, the central filter residue, equivalent to Asn653 of HsTPC2 is a glycine and, thus, renders the plant channel non-selective (*Guo et al., 2016*; *Guo et al., 2017b*).

The HsTPC2 ion conduction pore adopts both closed and open conformations in the ligand-bound state (*Figure 2B,C,D*). In the closed conformation, four pairs of residues consisting of Thr308s and Tyr312s from IS6 and Leu690s and Leu694s from IIS6, form the constriction points at the cytosolic side and prevent the passage of hydrated cations (*Figure 2B,C*). In the open state, these constriction-forming residues dilate and rotate away from the central axis, resulting in a much wider opening at the intracellular gate (*Figure 2B,C,D*). This pore opening and closing mechanics of HsTPC2 is highly similar to that of MmTPC1. The mechanism by which $PI(3,5)P_2$ regulates HsTPC2 gating will be further discussed later. Interestingly, we also observed two long stretches of density plugging the open gate in the open HsTPC2 structure (*Figure 2E*). This density likely comes from the GDN detergent used for sample preparation. Indeed, GDN can inhibit the HsTPC2 channel from the cytosolic side likely by functioning as an open-pore blocker (*Figure 2F*). A similar digitonin-like molecule occluding the open gate has also been reported in the structure of the voltage-gated sodium channel from electric eel (*Yan et al., 2017*).

## Voltage sensing domains in HsTPC2

Although the two voltage sensing domains (VSD) of HsTPC2 contain multiple positively charged arginines on the S4 helices, channel activation is achieved independently of the membrane potential. *Figure 3A* provides the sequence alignment of the S4 helices from TPCs and other canonical voltage-gated channels with the positions of the gating charge residues numbered from R1 to R5. VSD1 of HsTPC2 is structurally similar to that of MmTPC1. The IS4 helix contains three arginine residues at positions R2-R4 (Arg185, Arg188 and Arg191) with their charged side chains pointing towards the luminal surface (*Figure 3B*). However, several conserved features of a canonical voltage sensing domain are absent in HsTPC2 VSD1, thus rendering it voltage insensitive (*Figure 3B*): the acidic residue on S3 that is part of the gating charge transfer center is Tyr161; the conserved basic residue at the R5 position becomes Phe194; and IS4 forms a short regular alpha helix instead of a $3_{10}$ helix.

With two arginines at positions R4 (Arg554) and R5 (Arg557) on IIS4, VSD2 of HsTPC2 preserves some key features of a canonical voltage sensor, including the formation of a $3_{10}$ helix in a part of IIS4 and the presence of the conserved gating charge transfer center surrounded by Tyr480, Glu483 and Asp505 (*Figure 3C*). These structural features are highly similar to VSD2 of MmTPC1, which contributes to the voltage-dependent gating of TPC1, yet HsTPC2 is voltage independent (*Figure 3D*). One key difference between the VSD2 of these two channels is that the R3 position of HsTPC2 is an isoleucine instead of an arginine, which contributes to the loss of voltage-dependence in TPC2 channels (*Figure 3A,C*). In two recent studies, replacing the R3 arginine of mammalian TPC1 with Gln or Ile can yield a voltage-independent channel that can be activated solely by $PI(3,5)P_2$ – analogous to TPC2 (*Cang et al., 2014*; *She et al., 2018*). Conversely, we can also convert HsTPC2 to a voltage-dependent channel by introducing an arginine to the R3 position (Ile551Arg mutant), which yields a mutant channel whose activation requires both $PI(3,5)P_2$ and a positive membrane potential (*Figure 3E*). Thus, the presence or absence of an arginine at the R3 position determines the voltage dependence of mammalian TPC channels. In HsTPC2, the gating charge transfer center is occupied by Arg554 at the R4 position (*Figure 3C*). By introducing an R3 arginine in the Ile551Arg mutant, we envision that hyperpolarized (negative) membrane potential can drive a downward sliding motion of the IIS4 helix by one helical turn with the R3 arginine positioned at the gating charge transfer. As will be discussed later, this movement would occlude the space necessary for the outward movement of the IIS4-S5 linker that is associated with pore opening and, thereby, results in a voltage-dependent inhibition of channel gating similar to the voltage gating mechanism of MmTPC1 (*She et al., 2018*).

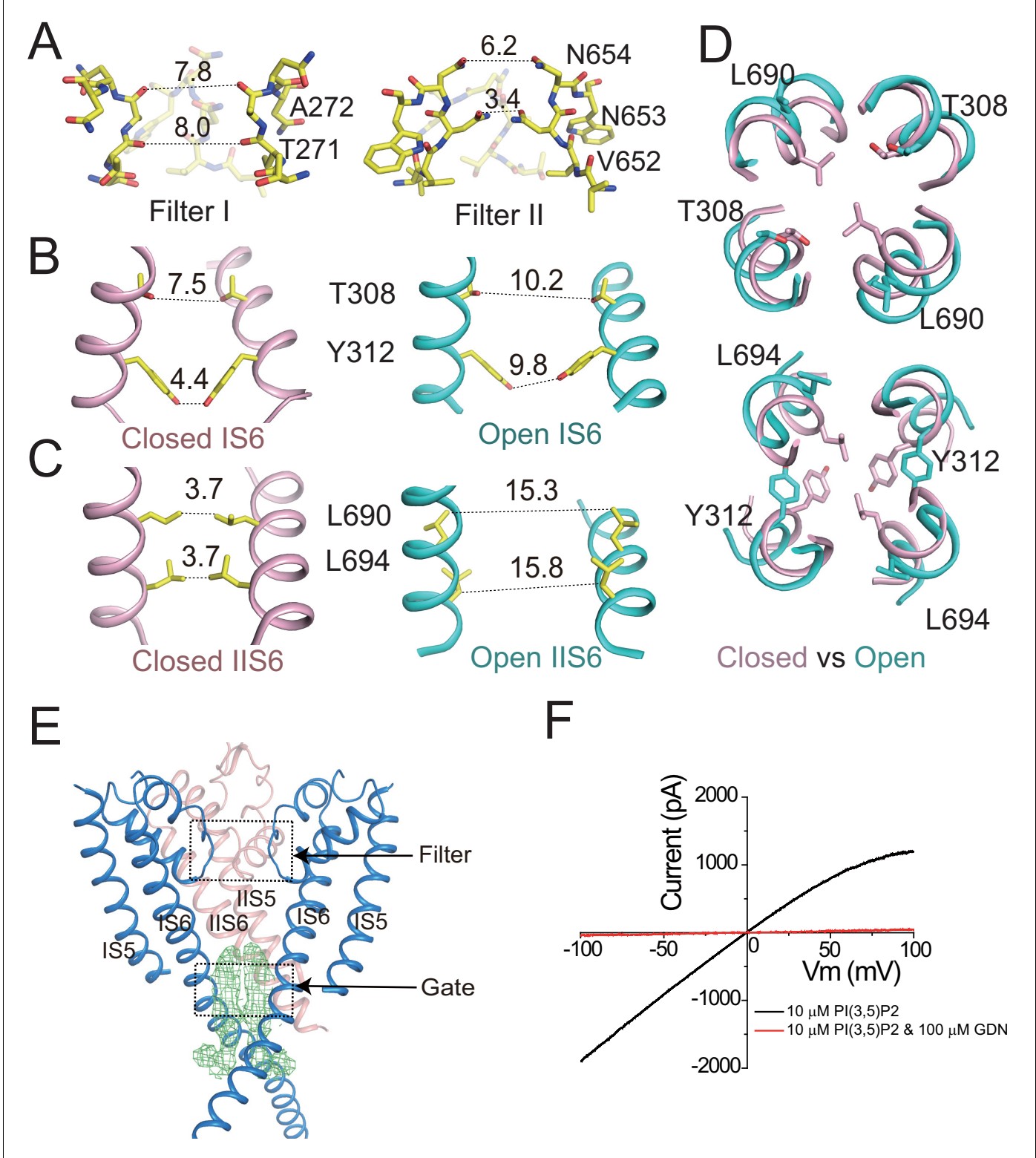

**Figure 2.** Ion conduction pore of HsTPC2. (**A**) The selectivity filter formed by filter I and filter II with the front subunit removed for clarity. (**B and C**) Side view of the bundle crossing formed by IS6 (**B**) and IIS6 (**C**) in the PI(3,5)P₂-bound closed (pink) and PI(3,5)P₂-bound open (cyan) states. Numbers are cross distances (in Å) at the constriction points. (**D**) Structural comparison of the cytosolic gate between the closed and open states viewed from the cytosolic side in two sections: Thr308/Leu690 (top) and Tyr312/Leu694 (below). (**E**) Side view of the EM density (green mesh) in the cytosolic gate of PI

*Figure 2 continued*

(3,5)P$_2$-bound open structure. The filter region and the cytosolic gate are boxed. (F) Sample *I–V* curves of HsTPC2 recorded using inside-out patches with or without GDN in bath solutions. Data shown in (F) were repeated five times independently with similar results.

DOI: https://doi.org/10.7554/eLife.45222.008

## PI(3,5)P$_2$ Binding in HsTPC2

The ligand affinity and specificity of HsTPC2 were measured in excised patches. The PI(3,5)P$_2$ concentration-dependent activation of the channel yielded an EC$_{50}$ of about 480 nM (*Figure 4A* and *Figure 4—figure supplement 1A*) similar to that measured in whole lysosome patch (*Wang et al., 2012*). In contrast, PI(4,5)P$_2$, PI(3,4)P$_2$, PI(3)P and PI(5)P have no obvious effect on channel activation, illustrating the high ligand specificity of HsTPC2 (*Figure 4B* and *Figure 4—figure supplement 1B*).

The PI(3,5)P$_2$ ligand can be clearly defined from the cryo-EM maps of the ligand-bound structures (*Figure 1—figure supplement 4C,D*). Similar to that in MmTPC1, PI(3,5)P$_2$ is situated at the junction formed by IS3, IS4, and the IS4-S5 linker of the 6-TM I (*Figure 4C,D*). The inositol 1,3,5-trisphosphate head group of PI(3,5)P$_2$ interacts with multiple basic residues and defines most of the ligand-protein interactions (*Figure 4C,D*), while the acyl chains of PI(3,5)P$_2$ insert upright into the membrane. One notable difference between HsTPC2 and MmTPC1 is that most of the basic residues involved in PI(3,5)P$_2$ binding, particularly those on the IS4-S5 linker, are predominantly lysines in HsTPC2 instead of arginines as in MmTPC1 (*Figure 4C,D* and *Figure 1—figure supplement 1*). Mutations of these lysine residues (Lys203, Lys204 and Lys207) have profound effect on PI(3,5)P$_2$ activation (*Figure 4E,F* and *Figure 4—figure supplement 1C,D,E*). Residues on IS6 also participate in PI(3,5)P$_2$ interactions in a state-dependent manner. In the ligand-bound, closed state, Arg329 forms a salt bridge with the C3-phosphate (*Figure 4C*), while in the ligand-bound open form, Arg329 forms salt bridges with both the C3- and C5-phosphates as well as a hydrogen bond with the C4 hydroxyl group; additionally, Ser322 forms a hydrogen bond with the C5-phosphate only in the open conformation (*Figure 4D*). This state-dependent PI(3,5)P$_2$ interaction with IS6 is associated with the IS6 conformational change that causes the opening and closing of the pore and will be discussed below. Between the two IS6 residues that interact with PI(3,5)P$_2$ in the open state, Arg329 appears to play the determinant role in channel gating as its mutation (Arg329Ala) almost completely abolishes channel activity whereas the Ser322Ala mutation reduces the channel activity to a lesser extent (*Figure 4E,F* and *Figure 4—figure supplement 1C,D,E*). This is distinct from MmTPC1 where a single lysine (Lys331) on IS6 at the position equivalent to Ser322 of HsTPC2 interacts with PI(3,5)P$_2$ and determines the ligand-dependent gating of MmTPC1 (*She et al., 2018*). It is worth noting that our structural observation of PI(3,5)P$_2$ binding in HsTPC2 matches quite well with the recent molecular dynamic simulation study that predicted the lipid binding site based on the structure of PI(3,5)P$_2$-insensitive plant TPC1 (*Kirsch et al., 2018*).

## PI(3,5)P$_2$-dependent gating mechanism in HsTPC2

The structures of HsTPC2 in three different states – apo closed, ligand-bound closed, and ligand-bound open – allowed us to elucidate the structural basis of the channel's PI(3,5)P$_2$-dependent gating mechanism (*Figure 5*). The apo structure of HsTPC2 is virtually identical to the ligand-bound closed state with the exception being the absence of PI(3,5)P$_2$ density (*Figure 5—figure supplement 1*), suggesting that the ligand binding pocket can readily accommodate PI(3,5)P$_2$ without undergoing any conformational change. Pore opening and closing is triggered by a conformational change at the IS6 helix. In the closed state, the IS6 helix breaks into two halves at Gly317, just below the cytosolic gate, while in the open state, IS6 becomes a long continuous helix (*Figure 4C,D* and *Figure 5A*). This conformational change is likely initiated by the interaction between Arg329 and PI(3,5)P$_2$. Upon ligand binding, the Arg329 side chain adopts an extended configuration to form salt bridges with the PI(3,5)P$_2$ phosphate groups (*Figure 5A*); this interaction provides an outward force to pull the IS6 helix into the straight conformation; meanwhile, Ser322 forms a hydrogen bond with the C5-phosphate of PI(3,5)P$_2$ to help stabilize the IS6 helix in the open conformation. The fact that we observed both closed and open conformations in the ligand-bound structure suggests that HsTPC2 toggles in dynamic equilibrium between these two states in the presence of PI(3,5)P$_2$ (*Figure 5C*).

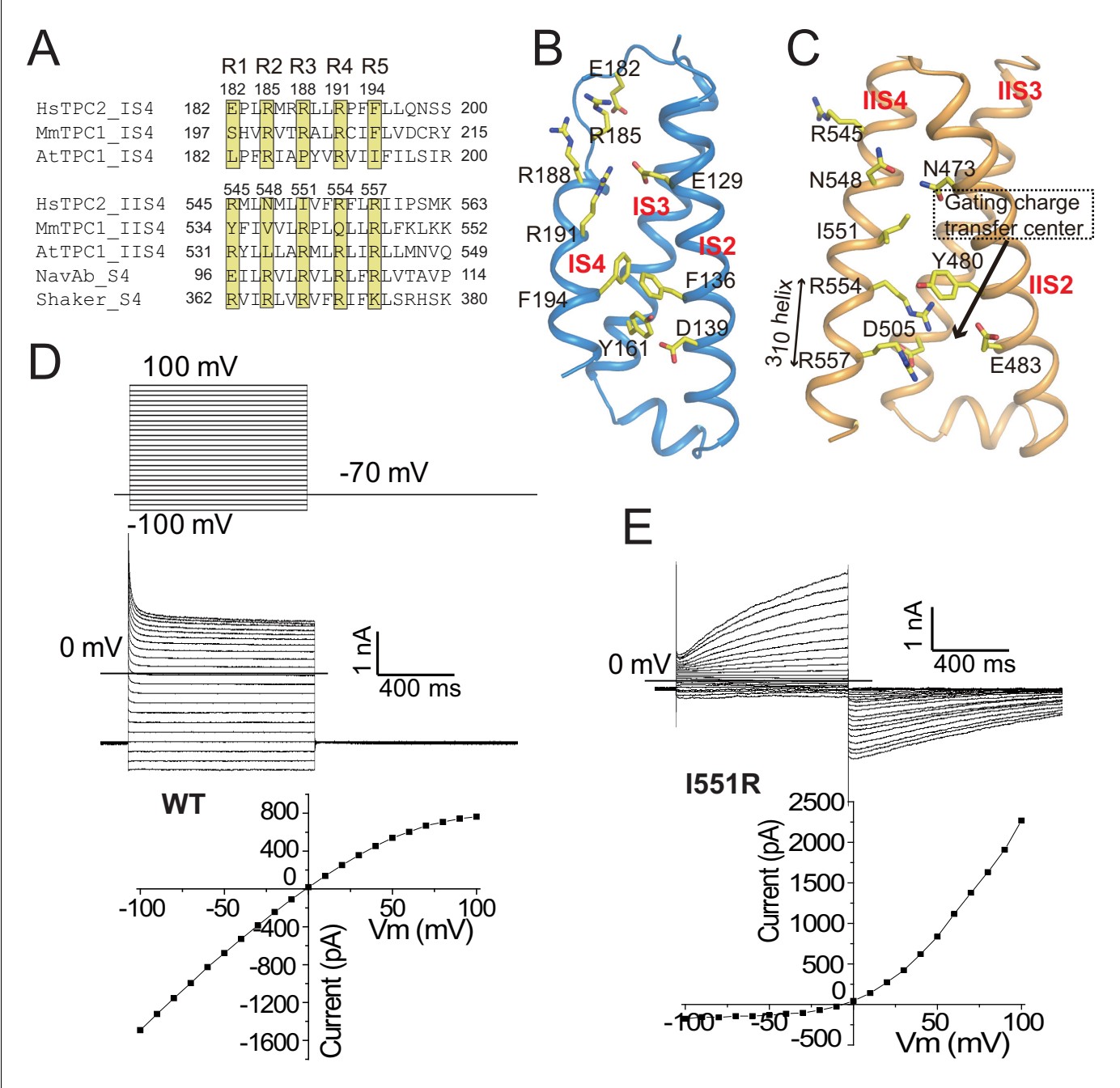

**Figure 3.** The voltage-sensing domains. (**A**) Partial S4 sequence alignment and arginine registry. Residues at the R1-R5 positions on IS4 and IIS4 of HsTPC2 are numbered. (**B**) Side view of VSD1 with IS1 omitted for clarity. (**C**) Side view of VSD2 with IIS1 omitted for clarity. (**D and E**) Sample traces and I-V curves of voltage activation of HsTPC2 (**D**) and its IIS4 arginine mutation Ile551Arg (**E**) recorded in the whole cell configuration with 10 µM PI(3,5) $P_2$ in the pipette. Sample I-V curves were obtained from the currents at the end of the activation voltage steps. Data shown in (**D**) and (**E**) were repeated five times independently with similar results.

DOI: https://doi.org/10.7554/eLife.45222.009

The movement of IS6 from the closed to open state results in the outward dilation and rotation of gating residues (Tyr312 and Thr308) at the bundle crossing (*Figure 2D*). To accommodate the PI(3,5)$P_2$-induced conformational change at IS6, the IIS6 pair of helices undergoes a concurrent movement to give rise to a similar outward dilation and rotation of the helix and, consequently, a similar motion of the IIS6 gating residues (Leu690 and Leu694) (*Figure 2D*). The concurrent motion at IIS6

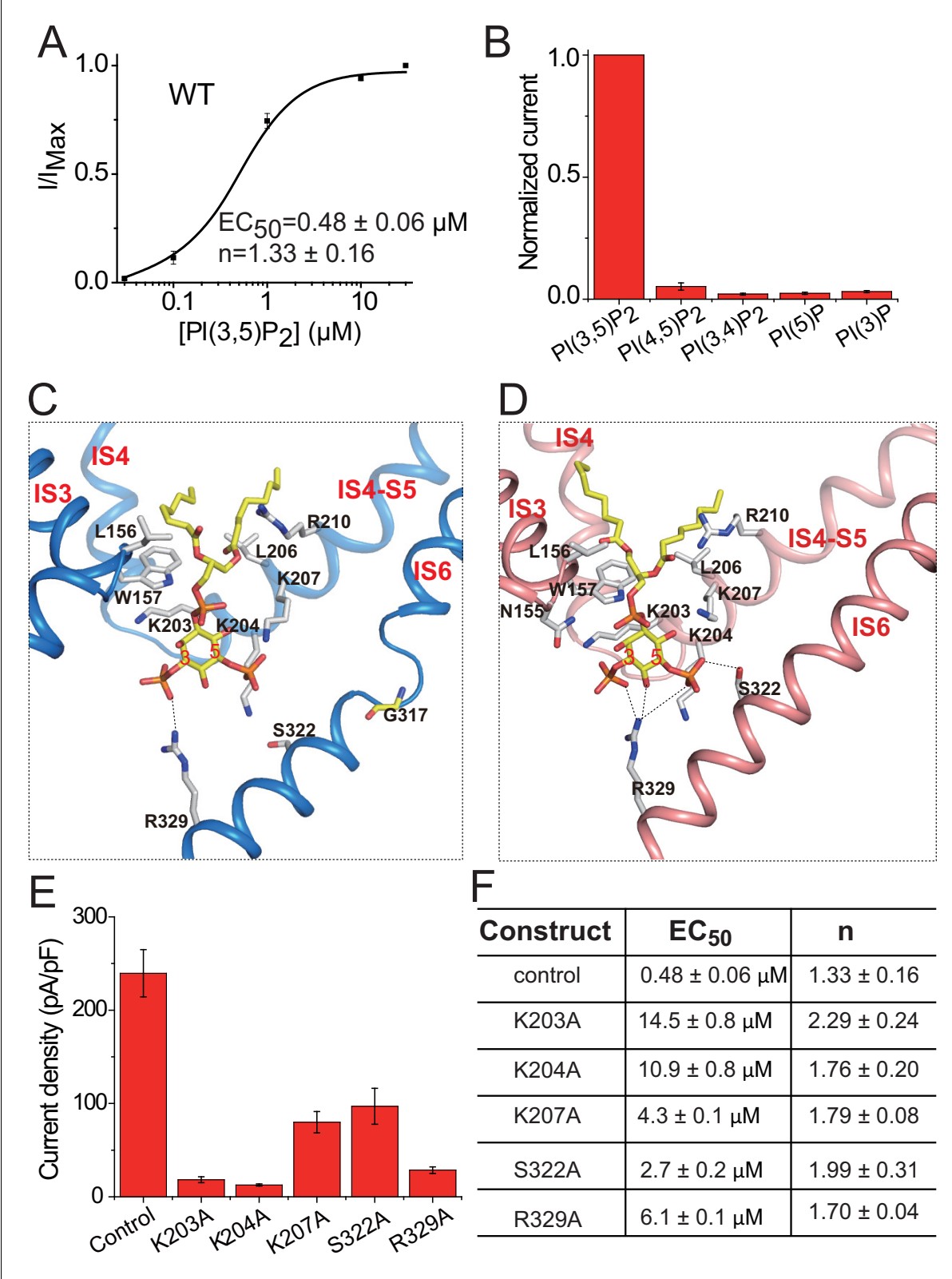

**Figure 4.** PI(3,5)P$_2$ binding in HsTPC2. (**A**) Concentration-dependent PI(3,5)P$_2$ activation of HsTPC2 at −100 mV. Curves are least square fits to the Hill equation. Data points are mean ± s.e.m. (n = 5 independent experiments). (**B**) Ligand specificity of HsTPC2. Normalized currents of HsTPC2 measured at −100 mV in inside-out patches with the presence of various phosphatidylinositol lipids at 10 μM. (**C and D**) Ligand binding site in 6-TM I of HsTPC2 both in the PI(3,5)P$_2$-bound closed (**C**) and open (**D**) states. (**E**) Current density of mutations at the PI(3,5)P$_2$-binding site measured at −100 mV by whole

*Figure 4 continued on next page*

*Figure 4 continued*

cell recordings with 100 µM PI(3,5)P$_2$. All mutants were generated on the background of the Leu11Ala/Leu12Ala mutant, which was used as the wild-type control. Data in (**B**) and (**E**) are shown as mean ±s.e.m. ($n \geq 6$ independent experiments). (**F**) Measured EC$_{50}$ and Hill coefficient values for the mutants. Data are mean ±s.e.m. ($n \geq 5$ independent experiments). Source data for *Figure 4A,B,E and F* are available in *Figure 4—source data 1*.

DOI: https://doi.org/10.7554/eLife.45222.010

The following source data and figure supplement are available for figure 4:

**Source data 1.** Source data for *Figure 4A, B, E and F*.

DOI: https://doi.org/10.7554/eLife.45222.012

**Figure supplement 1.** PI(3,5)P$_2$ binding in HsTPC2.

DOI: https://doi.org/10.7554/eLife.45222.011

hinges around the five-residue π-helix that is situated in the middle of IIS6 and propagated to a larger motion at the C-terminal end (*Figure 5B*). The presence of π-helix on S6 as a gating hinge has also been observed in some other tetrameric cation channels such as TRP channels (*Chen et al., 2017*; *Hughes et al., 2018*). Being tightly packed with the C-terminal part of IIS6, the IIS4-S5 linker has to swing outward along with IIS6. Interestingly, in the closed state, IIS4 and the IIS4-S5 linker is connected by a five-residue loop; upon channel opening, the loop undergoes structural rearrangement and the three residues preceding the IIS4-S5 linker helix are restructured to be part of the linker helix (*Figure 5B*). This restructuring allows the IIS4-S5 linker to swing outward upon channel opening without inducing movement in IIS4. Thus, the loop between IIS4 and the IIS4-S5 linker provides the necessary space for the linker motion that is associated with channel opening.

The structural change at the loop also explains the voltage-dependent gating of the Ile551Arg mutation in HsTPC2. The downward sliding motion of the IIS4 helix in the mutant channel – driven by membrane hyperpolarization – would occlude the space necessary for the IIS4-S5 linker's movement and, thereby, inhibit movement in IIS6 and channel opening (*Figure 5B*). In essence, this voltage-gating mechanism is similar to that of MmTPC1; that is, IIS4 movement in response to a change in membrane potential generates (depolarization) or occludes (hyperpolarization) the space necessary for the movement of the IIS6 and IIS4-S5 linker that is associated with PI(3,5)P$_2$-induced channel opening (*She et al., 2018*). In other words, voltage activation does not directly activate TPC channels but, instead, removes the inhibition exerted by the IIS4-S5 linker and allows the channel to be activated by PI(3,5)P$_2$.

## Conclusion

In this study, we reveal the structural mechanism of how PI(3,5)P$_2$ activates human TPC2. Phosphoinositide is a key signaling lipid that plays essential roles in many cellular signaling processes (*Di Paolo and De Camilli, 2006*; *Falkenburger et al., 2010*), and there are growing examples of ion channels and transporters whose functions are directly regulated by specific phosphoinositide (*Hilgemann et al., 2001*). While structural understanding of PIP$_2$ regulation of ion channels have been well-studied on the 2-TM inward rectifier channels (*Hansen et al., 2011*; *Whorton and MacKinnon, 2011*), little is known about the structural basis of PIP$_2$ activation or modulation of the more dominant 6-TM tetrameric cation channels. Our current study of HsTPC2, along with MmTPC1 in a previous study (*She et al., 2018*), provides crucial structural insights into the location of phosphoinositide binding, lipid isoform specificity, and mechanism of lipid activation. In addition to the TPC channel family, the lysosomal TRPML channel is also activated by PI(3,5)P$_2$ (*Dong et al., 2010*; *Feng et al., 2014*). However, the TRPML ligand-gating mechanism is likely to be different from that of TPC because the PI(3,5)P$_2$ binding occurs at the N-terminal pre-S1 region in TRPML – distal from the ion conduction pore (*Chen et al., 2017*; *Dong et al., 2010*). Furthermore, while PI(4,5)P$_2$ does not bind TPC1, it can competitively bind to TRPML and inhibits channel activity (*Zhang et al., 2012*). Several other members in the TRP channel family are known to be regulated by the plasma membrane specific PI(4,5)P$_2$. The PI(4,5)P$_2$ binding site in TRPV5 is analogous to the observed PI(3,5)P$_2$ binding site in TPC (*Hughes et al., 2018*). The activation of TRPM4 is modulated by PI(4,5)P$_2$, but the lipid binding site remains elusive (*Guo et al., 2017a*). Thus, various isoforms of phosphoinositide can impose diverse actions on a wide range of channels at different cellular locations and, thereby, generate a broad range of signaling responses.

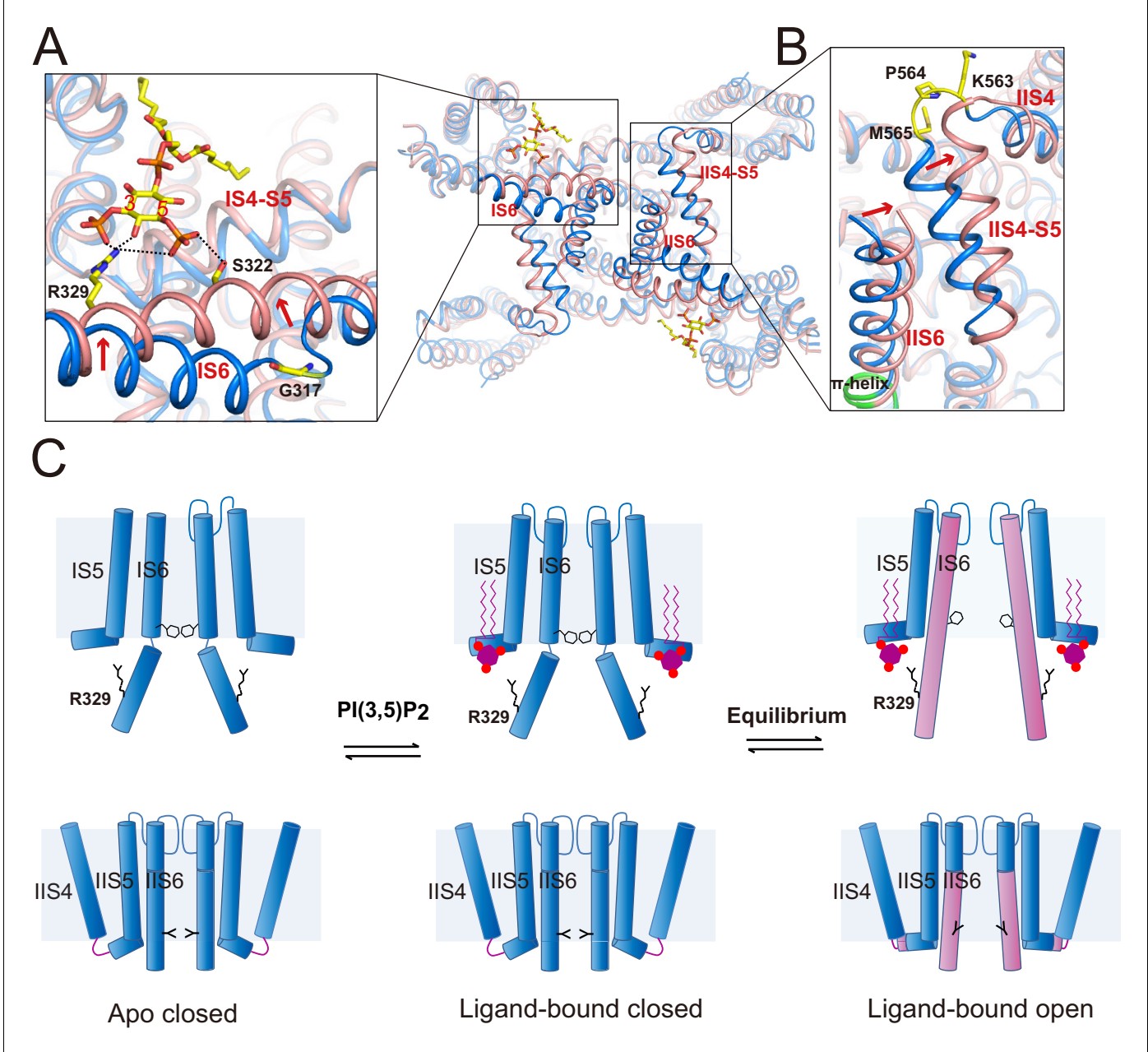

**Figure 5.** Gating mechanism for HsTPC2. (**A and B**) Structural comparison between the PI(3,5)P$_2$-bound open (pink) and closed (blue) HsTPC2 with zoomed-in views of the structural changes at IS6 (**A**) and IIS(6)/IIS4-S5 (**B**) regions. Arrows indicate structural movements. Key gating residues are shown as yellow sticks. The π-helix in IIS6 is shown in green. (**C**) Working model for PI(3,5)P$_2$ activation of HsTPC2.

DOI: https://doi.org/10.7554/eLife.45222.013

The following figure supplement is available for figure 5:

**Figure supplement 1.** PI(3,5)P$_2$ binding site in HsTPC2.

DOI: https://doi.org/10.7554/eLife.45222.014

# Materials and methods

**Key resources table**

| Reagent type (species) or resource | Designation | Source or reference | Identifiers | Additional information |
|---|---|---|---|---|
| Gene (*Human sapiens*) | TPC2 | DOI: 10.1038/nature08030 | NCBI: AY029200 | |
| Cell line (*Homo sapiens*) | Freestyle 293 F | Thermo Fisher Scientific | RRID:CVCL_D603 | |
| Cell line (*Spodoptera frugiperda*) | Sf9 | Thermo Fisher Scientific | RRID:CVCL_0549 | |
| Chemical compound | PI(3,5)P$_2$ diC8 | Echelon | Catalog No.: P-3508 | |
| Recombinant DNA reagent | pEZT-BM | DOI: 10.1016/j.str.2016.03.004 | Addgene: 74099 | |
| Software, algorithm | MotionCor2 | DOI: 10.1038/nmeth.4193 | | |
| Software, algorithm | GCTF | DOI: 10.1016/j.jsb.2015.11.003 | | |
| Software, algorithm | RELION2 | DOI: 10.7554/eLife.18722 | | |
| Software, algorithm | Coot | DOI: 10.1107/S0907444 910007493 | RRID:SCR_014222 | |
| Software, algorithm | Phenix | DOI: 10.1107/S2059 798318006551 | RRID:SCR_014224 | |
| Software, algorithm | PyMOL | Schrödinger | RRID:SCR_000305 | |
| Software, algorithm | Chimera | UCSF | RRID:SCR_004097 | |

## Protein expression and purification

The Leu11Ala/Leu12Ala mutant of Human TPC2 (HsTPC2, NCBI accession: AY029200) containing an N-terminal 8x His tag followed by a GFP tag and a thrombin cleavage site was cloned into a pEZT-BM vector (**Morales-Perez et al., 2016**) and heterologously expressed in Freestyle 293F cells (Thermo Fisher Scientific) using the BacMam system (Thermo Fisher Scientific). The Leu11Ala and Leu12Ala mutations promote channel expression and trafficking to the plasma membrane of the HEK293 cell. The baculovirus was generated in Sf9 cells (Thermo Fisher Scientific) following standard protocol and used to infect Freestyle 293F cells at a ratio of 1:40 (virus:cells, v:v) and supplemented with 10 mM sodium butyrate to boost protein expression. Cells were cultured in suspension at 37°C for 48 hr and harvested by centrifugation at 3000 x g. All purification procedures were carried out at 4°C. The cell pellet was re-suspended in buffer A (20 mM Tris, pH 8.0, 150 mM NaCl) supplemented with a protease inhibitor cocktail (containing 2 µg/ml DNAse, 0.5 µg/ml pepstatin, 2 µg/ml leupeptin, and 1 µg/ml aprotinin and 0.1 mM PMSF) and homogenized by sonication on ice. HsTPC2 was extracted with 1% (w:v) n-Dodecyl-β-D-Maltopyranoside (DDM) (Anatrace) supplemented with 0.2% (w:v) cholesteryl hemisuccinate (CHS, Sigma Aldrich) by gentle agitation for 2 hr. After extraction, the supernatant was collected after a 60 min centrifugation at 20,000 x g and incubated with Ni-NTA resin (Qiagen) using gentle agitation. After 2 hr, the resin was collected on a disposable gravity column (Bio-Rad). The resin was washed with buffer B (20 mM Tris, pH 8.0, 150 mM NaCl and 0.06% glycol-diosgenin (GDN, Anatrace) supplemented with 20 mM imidazole. The washed resin was left on column in buffer B and digested with thrombin (Roche) overnight. After thrombin digestion, the flow-through containing untagged HsTPC2 was collected, concentrated, and purified by size exclusion chromatography on a Superdex 200 column (GE Healthcare) pre-equilibrated with buffer B. The peak fraction was pooled and concentrated to 3.3 mg/ml for cryo-electron microscopy analysis. To obtain the phosphatidylinositol 3,5-bisphosphate (PI(3,5)P$_2$) bound structure, the protein sample was

supplemented with 0.5 mM PI(3,5)P$_2$ diC8 (Echelon) for 30 min on ice before EM grids preparation. HEK293 (CRL-1573) cell lines were purchased from and authenticated by the American Type Culture Collection (ATCC, Manassas, VA). Freestyle 293F cells (RRID:CVCL_D603) were purchased from and authenticated by Thermo Fisher Scientific. The cell lines were not tested for mycoplasma contamination as they were used for protein expression only.

## EM data acquisition

The cryo-EM grids were prepared by applying HsTPC2 (3.3 mg/ml, with or without 0.5 mM PI(3,5)P$_2$) to a glow-discharged Quantifoil R1.2/1.3 300-mesh gold holey carbon grid (Quantifoil, Micro Tools GmbH, Germany). Grids were blotted for 4.0 s under 100% humidity at 4°C before being plunged into liquid ethane using a Mark IV Vitrobot (FEI). Micrographs were acquired on a Titan Krios microscope (FEI) operated at 300 kV with a K2 Summit direct electron detector (Gatan), using a slit width of 20 eV on a GIF-Quantum energy filter. Images were recorded with EPU software (FEI) in super-resolution counting mode with a super resolution pixel size of 0.535 Å. The defocus range was set from −1.5 μm to −3 μm. Each micrograph was dose-fractionated to 30 frames under a dose rate of 4 e-/pixel/s, with a total exposure time of 15 s, resulting in a total dose of about 50 e-/Å$^2$.

## Image processing

Micrographs were motion corrected and binned two fold (yielding a pixel size of 1.07 Å/pixel) with MotionCor2 (*Zheng et al., 2017*). The CTF parameters of the micrographs were estimated using the GCTF program (*Zhang, 2016*). All other steps of image processing were performed using RELION2.0 (*Kimanius et al., 2016*). Initially, ~1000 particles were manually picked from a few micrographs. Class averages representing projections of HsTPC2 in different orientations were selected from the 2D classification of the manually picked particles and used as templates for automatic particle picking from the full dataset. 653,430 particles were picked from 2287 micrographs. The particles were extracted and binned three times (3.21 Å/pixel). After 2D classification, a total of 473,552 particles were finally selected for 3D classification using MmTPC1 structure (PDB accession number 6C9A) as the initial mode. The 3D classes showed good secondary structure features and were selected and re-extracted into the original pixel size of 1.07 Å. After 3D refinement with C2 symmetry imposed and particle polishing, the resulting 3D reconstructions from ~310 K particles showed a clear 2-fold symmetry with a resolution of 3.4 Å. We then performed a focused 3D classification with density subtraction (*Bai et al., 2015*). In this approach, only the density corresponding to the pore domain was kept in each particle image by subtracting the density for all other parts including the belt-like detergent density from the original particles. The subsequent 3D classification on the modified particles was carried out by applying a mask around the pore domain with all the particle orientations fixed at the value determined in the initial 3D refinement. After this round of classification, one class showing an open gate (~33K particles, ligand-bound open state) was selected and 3D refined to get an EM map of 3.7 Å, and two other classes showing a closed gate (~56K particles, ligand-bound closed state) were selected and 3D refined to get an EM map of 3.4 Å.

Two datasets were collected for the apo HsTPC2 sample. For dataset I, 101,354 particles were picked from 327 micrographs. After 2D classification, 86,560 particles were selected for 3D classification. For dataset II, 126,572 particles were picked from 368 micrographs. After 2D classification, 105,629 particles were selected for 3D classification. One class from 3D classification of dataset I (31, 973 particles) and two classes from 3D classification of dataset II (64, 388 particles) were combined for 3D auto-refinement and particle polishing, which resulted in a map with an overall resolution of 3.5 Å.

All resolutions were estimated by applying a soft mask around the protein density and the gold-standard Fourier shell correlation (FSC) = 0.143 criterion. ResMap was used to calculate the local resolution map (*Kucukelbir et al., 2014*).

## Model building, refinement, and validation

Model buildings were conducted in Coot (*Emsley et al., 2010*) and the MmTPC1 structure (PDB accession number 6C9A) was used as the reference. Amino acid assignments were also confirmed by the clearly defined densities for bulky residues (Phe, Trp, Tyr, and Arg). Real space model refinement (*Afonine et al., 2018b*) and validation (*Afonine et al., 2018a*) were performed in Phenix.

Residues 1–38, 241–251, 347–353, 526–538, 609–619 and 702–752 are disordered and not modeled in all states. The statistic of the models' geometries was generated using MolProbity (*Chen et al., 2010*). All figures were prepared using the software PyMol (RRID:SCR_000305, The PyMOL Molecular Graphics System, Version 1.8 Schrödinger, LLC.) or Chimera (*Pettersen et al., 2004*).

## Electrophysiology

In human TPC2, the Leu11Ala and Leu12Ala mutations at the N-terminal targeting sequence have been shown to promote channel expression and trafficking to the plasma membrane of the HEK293 cell, allowing for channel activity measurement using patch clamp (*Brailoiu et al., 2010*; *Jha et al., 2014*). Therefore the Leu11Ala/Ile12Ala mutant was used and considered as the wild type channel in all our recordings. All other mutations in our experiments were generated on the background of this plasma membrane-targeting HsTPC2. With the channels targeted to the plasma membrane, the extracellular side is equivalent to the luminal side of HsTPC2 in endo/lysosomes. HsTPC2 and its mutants were cloned into pCS2 vector. About 2 μg of the plasmid containing the C-terminal GFP-tagged mouse TPC2 or its mutants was transfected into HEK293 cells grown in a six-well tissue culture dish using Lipofectamine 2000 (Thermo Fisher Scientific). 48 hr after transfection, cells were dissociated by trypsin treatment and kept in complete serum-containing media and re-plated on a 35 mm tissue culture dish and kept in a tissue culture incubator until recording. Similar transfections were done for expression test and subcellular localization experiments with the exception being 0.5 ug of the plasma membrane maker KRas GTPase tail tagged with mCherry was co-transfected with the HsTPC2 mutants for the subcellular localization experiments.

Patch-clamp recordings in the whole-cell configuration or inside-out configuration (excised patch) were employed to measure channel activity. The standard intracellular solution contained (in mM): 145 Sodium methanesulfonate (Na-MS), 5 NaCl, 1 EGTA, 10 HEPES buffered with Tris, pH = 7.4. The extracellular solution contained (in mM): 145 Na-MS, 5 NaCl, 1 $MgCl_2$, 1 $CaCl_2$, 10 HEPES buffered with Tris, pH = 7.4. Various concentrations of $PI(3,5)P_2$ as specified in each experiment were added to the intracellular solutions to activate the channel. For patches in the whole-cell configuration, the intracellular and extracellular solutions are defined as that in the pipette and the bath, respectively; the solution arrangement is reversed for the inside-out patch recordings. The lipid ligands used in our studies are phosphatidylinositol-3,4-bisphosphate diC8 ($PI(3,4)P_2$ diC8), phosphatidylinositol-3,5-bisphosphate diC8 ($PI(3,5)P_2$ diC8), phosphatidylinositol-4,5-bisphosphate diC8 ($PI(4,5)P_2$ diC8), phosphatidylinositol-3-phosphate diC8 (PI(3)P diC8), and phosphatidylinositol-5-sphosphate diC8 (PI(5)P diC8) (Echelon).

Electrophysiology data were acquired using an AxoPatch 200B amplifier (Molecular Devices) and a low-pass analogue filter set to 1 kHz. The current signal was sampled at a rate of 20 kHz using a Digidata 1322A digitizer (Molecular Devices) and further analyzed with pClamp9 software (Molecular Devices). Patch pipettes were pulled from borosilicate glass (Harvard Apparatus) and heat polished to a resistance of 3–5 MΩ. After the patch pipette attached to the cell membrane, a giga seal (>10 GΩ) was formed by gentle suction. The whole-cell configuration was formed by short zap or suction to rupture the patch. The inside-out configuration was formed by pulling pipette away from the cell, and the pipette tip was exposed into the air for short time in some cases. All I-V curves of HsTPC2 and its mutants were obtained using voltage pulses ramp from −100 to +100 mV over 800 ms duration with the exception being the data for voltage activation shown in *Figure 3D and E*. In the voltage-dependent activation measurements of HsTPC2 and the HsTPC2 R551I mutant, the membrane was stepped from the holding potential (−70 mV) to various testing potentials (−100 mV to +100 mV) for 1 s and then stepped to −70 mV. The current at the end of each testing potential was used to plot the I-V curve. All electrophysiological recordings were repeated at least five times using different patches. All data points are mean ± s.e.m. (n ≥ 5).

## Data availability

The cryo-EM density maps of HsTPC2 have been deposited in the Electron Microscopy Data Bank under accession numbers EMD-0478 for the apo state, EMD-0477 for the $PI(3,5)P_2$-bound open state and EMD-0479 for the $PI(3,5)P_2$-bound closed state. Atomic coordinates have been deposited in the Protein Data Bank under accession numbers 6NQ1 for the apo state, 6NQ0 for the $PI(3,5)P_2$-bound open state and 6NQ2 for the $PI(3,5)P_2$-bound closed state.

## Acknowledgements

We thank N Nguyen for manuscript preparation, and Dr. M X Zhu at University of Texas Health Science Center at Houston for providing clones of animal TPC genes. Single particle cryo-EM data were collected at the University of Texas Southwestern Medical Center Cryo-EM Facility that is funded by the CPRIT Core Facility Support Award RP170644. This work was supported in part by the Howard Hughes Medical Institute (YJ) and by grants from the National Institute of Health (GM079179 to YJ) and the Welch Foundation (Grant I-1578 to YJ, grant I-1944-20180324 to XB). XB is supported by the Cancer Prevention and Research Initiative of Texas and Virginia Murchison Linthicum Scholar in Medical Research fund.

## Additional information

### Funding

| Funder | Grant reference number | Author |
|---|---|---|
| Cancer Prevention and Research Institute of Texas | | Xiao-chen Bai |
| University of Texas Southwestern Medical Center | Virginia Murchison Linthicum Scholar in Medical Research fund | Xiao-chen Bai |
| Howard Hughes Medical Institute | | Youxing Jiang |
| National Institute of General Medical Sciences | GM079179 | Youxing Jiang |
| Welch Foundation | I-1578 | Youxing Jiang |
| Welch Foundation | I-1944-20180324 | Xiao-chen Bai |

The funders had no role in study design, data collection and interpretation, or the decision to submit the work for publication.

### Author contributions

Ji She, Conceptualization, Data curation, Formal analysis, Validation, Investigation, Visualization, Writing—original draft, Writing—review and editing, Prepared the samples, Performed data acquisition, image processing and structure determination, Participated in research design, data analysis, and manuscript preparation; Weizhong Zeng, Data curation, Formal analysis, Investigation, Visualization, Writing—review and editing, Performed electrophysiology, Participated in research design, data analysis, and manuscript preparation; Jiangtao Guo, Conceptualization, Resources, Data curation, Investigation, Writing—review and editing, prepared the samples, Performed data acquisition, image processing and structure determination, Participated in research design, data analysis, and manuscript preparation; Qingfeng Chen, Conceptualization, Data curation, Investigation, Writing—review and editing, prepared the samples, Performed data acquisition, image processing and structure determination, Participated in research design, data analysis, and manuscript preparation; Xiao-chen Bai, Data curation, Software, Formal analysis, Supervision, Validation, Investigation, Methodology, Writing—review and editing, Performed data acquisition, image processing and structure determination, Participated in research design, data analysis, and manuscript preparation; Youxing Jiang, Conceptualization, Resources, Data curation, Supervision, Funding acquisition, Visualization, Writing—original draft, Project administration, Writing—review and editing, Participated in research design, data analysis, and manuscript preparation

### Author ORCIDs

Ji She http://orcid.org/0000-0001-7006-6230
Youxing Jiang http://orcid.org/0000-0002-1874-0504

### Decision letter and Author response

Decision letter https://doi.org/10.7554/eLife.45222.029

Author response https://doi.org/10.7554/eLife.45222.030

## Additional files

### Supplementary files

• Transparent reporting form

DOI: https://doi.org/10.7554/eLife.45222.015

### Data availability

The cryo-EM density maps of the human TPC2 have been deposited in the Electron Microscopy Data Bank under accession numbers EMD-0478 for the apo state, EMD-0479 for the PI(3,5)P2-bound closed state and EMD-0477 for the PI(3,5)P2-bound open state. Atomic coordinates have been deposited in the Protein Data Bank under accession numbers 6NQ1 for the apo state, 6NQ2 for the PI(3,5)P2-bound closed state and 6NQ0 for the PI(3,5)P2-bound open state.

The following datasets were generated:

| Author(s) | Year | Dataset title | Dataset URL | Database and Identifier |
|---|---|---|---|---|
| She J, Zeng W, Guo J, Chen Q, Bai X-c, Jiang Y | 2019 | Atomic coordinates of the human TPC2 (PI(3,5)P2-bound open state) | http://www.rcsb.org/structure/6NQ0 | Protein Data Bank, 6NQ0 |
| She J, Zeng W, Guo J, Chen Q, Bai X-c, Jiang Y | 2019 | Cryo-EM density map of the human TPC2 (apo state) | http://www.ebi.ac.uk/pdbe/entry/emdb/EMD-0478 | Electron Microscopy Data Bank, EMD-0478 |
| She J, Zeng W, Guo J, Chen Q, Bai X-c, Jiang Y | 2019 | Cryo-EM density map of the human TPC2 (PI(3,5)P2-bound closed state) | http://www.ebi.ac.uk/pdbe/entry/emdb/EMD-0479 | Electron Microscopy Data Bank, EMD-0479 |
| She J, Zeng W, Guo J, Chen Q, Bai X-c, Jiang Y | 2019 | Cryo-EM density map of the human TPC2 (PI(3,5)P2-bound open state) | http://www.ebi.ac.uk/pdbe/entry/emdb/EMD-0477 | Electron Microscopy Data Bank, EMD-0477 |
| She J, Zeng W, Guo J, Chen Q | 2019 | Atomic coordinates of the human TPC2 (apo state) | http://www.rcsb.org/structure/6NQ1 | Protein Data Bank, 6NQ1 |
| She J, Zeng W, Guo J, Chen Q, Bai X-c, Jiang Y | 2019 | Atomic coordinates of the human TPC2 (PI(3,5)P2-bound closed state) | http://www.rcsb.org/structure/6NQ2 | Protein Data Bank, 6NQ2 |

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
