## [Decision Letter]

Thank you for submitting your article "Structural mechanisms of phospholipid activation of the human TPC2 channel" for consideration by *eLife*. Your article has been reviewed by Richard Aldrich as the Senior Editor, a Reviewing Editor, and three reviewers. The following individuals involved in review of your submission have agreed to reveal their identity: Haoxing Xu (Reviewer #1); Alexander Sobolevsky (Reviewer #2); Vera Moiseenkova-Bell (Reviewer #3).

The reviewers have discussed the reviews with one another and the Reviewing Editor has drafted this decision to help you prepare a revised submission.

Summary:

TPC channels TPC1 and TPC2 are present in endolysosomal membranes and implicated in various physiological processes, including hair pigmentation, blood vessel formation and lipid metabolism. Determining structural bases of their architecture and gating is important not only for understanding the role of these channels in physiological processes but may also help developing new therapeutic strategies. She et al. present cryo-EM structures of human TPC2 in the apo and agonist (PIP_2_)-bound closed and open states. The same group has previously solved structures of TPC1 from *Arabidopsis thaliana* and mouse. While the structures and gating mechanisms of TPC1 and TPC2 are similar, there are significant subtype-specific differences, including the lack of TPC2 modulation by the membrane voltage, which undoubtedly warrant publication of these new results in *eLife*. The data presented is of high quality and the interpretations are just and fair.

Essential revisions:

We only have one essential suggestion: Density comparison analysis suggested that K203, K204 K207, S322, and R329 form the PI(3,5)P_2_-binding site(s), which was supported by the electrophysiology results on single mutations. Only EC_50_ values were provided. It would be most helpful if the hill slope (n) values are also provided for wild-type and mutant channels. If these mutations are indeed affect ligand-binding, but not gating or gating-binding coupling, it is likely that the n values are reduced. Of course it would be more convincing if the PI(3,5)P_2_ activation effect was abolished by double or triple mutations of K203/K204/K207, yet activation by another TPC agonist remains intact.

---

## [Author Response]

Essential revisions:We only have one essential suggestion: Density comparison analysis suggested that K203, K204 K207, S322, and R329 form the PI(3,5)P_2_-binding site(s), which was supported by the electrophysiology results on single mutations. Only EC_50_ values were provided. It would be most helpful if the hill slope (n) values are also provided for wild-type and mutant channels. If these mutations are indeed affect ligand-binding, but not gating or gating-binding coupling, it is likely that the n values are reduced. Of course it would be more convincing if the PI(3,5)P_2_ activation effect was abolished by double or triple mutations of K203/K204/K207, yet activation by another TPC agonist remains intact.

The Hill coefficient (n) values have been added in the revision (Figure 4F) as suggested. We actually noticed a slight increase in the Hill coefficient values for all the mutants. It is possible that the mutations also affect the coupling between gating and ligand binding. We did not test any double or triple mutations as single mutation at K203 or K204 already resulted in very small currents. We suspect that the double or triple mutations at K203/K204/K207 may not yield any measurable currents. We do not know of any agonist that can activate TPC channel independently of PI(3,5)P_2_.